# Joint Texture Search and Histogram Redistribution for Hyperspectral Image Quality Improvement

**DOI:** 10.3390/s23052731

**Published:** 2023-03-02

**Authors:** Bingliang Hu, Junyu Chen, Yihao Wang, Haiwei Li, Geng Zhang

**Affiliations:** 1Key Laboratory of Spectral Imaging Technology of Chinese Academy of Sciences, Xi’an Institute of Optics and Precision Mechanics of CAS, Xi’an 710119, China; 2University of Chinese Academy of Sciences, Beijing 100049, China

**Keywords:** 4D block aggregation, BM4D, histogram redistribution, hyperspectral quality enhancement

## Abstract

Due to optical noise, electrical noise, and compression error, data hyperspectral remote sensing equipment is inevitably contaminated by various noises, which seriously affect the applications of hyperspectral data. Therefore, it is of great significance to enhance hyperspectral imaging data quality. To guarantee the spectral accuracy during data processing, band-wise algorithms are not suitable for hyperspectral data. This paper proposes a quality enhancement algorithm based on texture search and histogram redistribution combined with denoising and contrast enhancement. Firstly, a texture-based search algorithm is proposed to improve the accuracy of denoising by improving the sparsity of 4D block matching clustering. Then, histogram redistribution and Poisson fusion are used to enhance spatial contrast while preserving spectral information. Synthesized noising data from public hyperspectral datasets are used to quantitatively evaluate the proposed algorithm, and multiple criteria are used to analyze the experimental results. At the same time, classification tasks were used to verify the quality of the enhanced data. The results show that the proposed algorithm is satisfactory for hyperspectral data quality improvement.

## 1. Introduction

In the past few decades, hyperspectral imaging has become a significant approach in many fields, and hyperspectral image (HSI) processing has attracted extensive attention [1,2]. For example, in urban planning, surveying and mapping, agriculture, forestry and disaster monitoring [3,4,5,6], the quantitative analysis capability of hyperspectral remote sensing imaging data provides the means to identify material properties deeply. However, due to the limitations of equipment hardware performance (optical and electrical noise), compression difficulties, and information loss during transmission, there are many influences on hyperspectral image data processing. Consequently, the visual effect and application value are seriously affected. Therefore, studying HSI quality improvement technology is one of the main issues to solve on the road to scientific and technological innovation.

During the past decade, there were relatively little research on hyperspectral quality improvement. Recently, more researchers focused on hyperspectral denoising [7,8,9]. With the development of artificial intelligence, researchers often use a neural network to de-noise hyperspectral data [10,11,12,13]. Despite this, in the actual test data, the denoising effect largely depends on the generalization performance of the training samples and the network. For newly obtained HSI, the quality needs improvement. In addition, few scholars studied the contrast enhancement of each band in HSI. On the other hand, in practical applications, high DN values (overexposed clouds) and low DN values (dark pixels, water bodies) with fewer pixels will lead to valuable objects in a narrow dynamic range [14,15,16]. Even with linear stretching, the visualizations are still low, significantly reducing customer perception. A better visualization requirement for HSI was proposed in the literature [17,18,19]. However, there is no further analysis on whether the spectral information is well preserved. The visualization effect is enhanced but spectral information is lost. It is not conducive to later data processing, such as classification and detection.

According to the research status of HSI degradation, an HSI quality improvement algorithm, combining denoising and contrast enhancement, is proposed. The main contributions of this paper are summarized as follows:(1)Considering the noise interference in degraded HSI, it is innovatively proposed to extract the edge features of 3D similar block aggregation to restore HSI’s spectral space features by optimizing the sparsity of similar blocks.(2)Aiming at the current situation of poor visual effect caused by low spatial contrast, the adaptive threshold constraint histogram contraction combined with Poisson fusion is innovatively proposed to improve global and local contrast.

The rest of the paper is organized as follows. In Section 2, a quality improvement algorithm for degraded HSI is proposed. The experiment results and discussion are presented and discussed in Section 3. Section 4 is a summarization of this paper.

## 2. Materials and Methods

As mentioned in the introduction, current research lacks a robust method to improve HSI quality (denoising and contrast enhancement). The Block Matching 4D (BM4D) algorithm [20,21] provides a relatively stable denoising model. However, its block matching is not optimal, which affects the denoising performance and may decrease the dynamic range. An HSI enhancement algorithm based on texture search and histogram redistribution is proposed to solve these problems. The process flow chart of the algorithm is shown in Figure 1.

Firstly, due to the redundancy of hyperspectral bands, the Minimum Noise Fraction (MNF) [22,23,24,25] dimension reduction algorithm is utilized to obtain a single-channel feature map. Secondly, the direction and amplitude of the gradient are calculated on the characteristic graph. The non-maximum suppression method enhances the edge information to obtain more accurate texture. Thirdly, the obtained precise texture image is divided into an edge and non-edge region. In the non-edge region, similar blocks are searched vertically and horizontally. However, in the edge region, similar blocks are searched along the edge direction (see Section 2.1). Additionally, the intermediate denoising results are obtained by initial 4D block grouping, 4D hard threshold filtering, primary aggregation, secondary 4D block estimation, Wiener filtering, and secondary aggregation. Finally, the hyperspectral contrast and spatial detail information are improved by histogram redistribution and Poisson fusion under the unchanged spectral characteristics (see Section 2.2).

### 2.1. Texture-Based 4D Block Aggregation

Traditional BM4D is along a local square area in the X and Y directions. Similar blocks are not optimal. Hyperspectral remote sensing data of collected scenes such as farmland, rivers, and mountains have clear edge texture information. A better similar block could be searched when moving along the scene texture. Therefore, the search method of BM4D similar blocks should be improved.

When the center of the reference block falls on the non-edge, search the fixed-size image blocks in the rectangular window in the horizontal and vertical directions to determine which image blocks are similar to the reference block.

When the center point of the reference block falls on the target edge pixel, cluster 4D-similar blocks along the gradient and directional characteristics of the edge. However, HSI contains noise in each band, making extracting the edge difficult. In order to obtain clear edge texture, use MNF to obtain the feature map first. Then, the Canny algorithm extracts and marks the target edge texture $\left\{S_{1},S_{2}\ldots S_{n}\right\}$. For the corresponding subset of curves, expand l distance along the normal direction of the curve $\left\{L_{1},L_{2}\ldots L_{n}\right\}$. Obtain the corresponding search area $\left\{S_{n}\times L_{n}\right\}$. Finally, the 3D cube with the smallest distance $L_{\left(RMSE\_SAM\right)}$ is calculated and integrated into a 4D matrix. The $L_{\left(RMSE\_SAM\right)}$ distance is as follows:(1)$$L_{\left(RMSE\_SAM\right)}=\lambda_{1}\sqrt{\frac{1}{m}\sum_{i=1}^{m}{\left(y_{i}-{\hat{y}}_{i}\right)}^{2}}+\lambda_{2}\sqrt{\frac{1}{m}\sum_{i=1}^{m}(cos^{-1}\frac{y_{i}^{T}{\hat{y}}_{i}}{{\left(y_{i}^{T}y_{i}\right)}^{1/2}{\left({\hat{y}}_{i}^{T}{\hat{y}}_{i}\right)}^{1/2}})},$$
where $y_{i}$ represents the DN value of the reference block at position *i*, and ${\hat{y}}_{i}$ represents the DN value of similar blocks corresponding to the reference block at position i.

### 2.2. Preserving Spectral Information for Contrast Enhancement

Most histograms are in a narrow range. Even with linear stretching, the contrast is still low. Therefore, remote sensing imaging often uses a 2% truncation stretch to enlarge the contrast. However, this approach loses much detail information. Based on the fact that the information entropy is maximum when the image histogram is uniformly distributed, try to achieve maximum information entropy and dynamic range by redistributing the corresponding hyperspectral histogram, while preserving the details of the image. The specific algorithm is as follows.

Firstly, process HSI data by band. Determine the convergence threshold $T_{band}=w*h/2^{N}$ of histogram redistribution and the initial search DN value $DN_{start}=\frac{1}{w*h}\sum_{j=1}^{w}\sum_{i=1}^{n}f_{b}\left(i,j\right)$ in each band.

Then, starting from the initial value $DN_{start}$, the searched DN value is accumulated along the positive and negative directions of the X-axis. Judge whether the current DN cumulative value is close to the convergence threshold $T_{band}$. If the accumulated DN value is close to the convergence threshold *T_band_*, record and update the current DN value. Otherwise, continue to search and accumulate DN values until all DN values in this band are processed.

Next, find the connected region of the merged pixels to be spatially related. Obtain the original DN value of its connected region and do the local remapping to restore more local contrast information. The local mapping is to map the spatially related pixels of the merged region back in sequence along the merged gray value. Stop the mapping when exceeding the maximum and minimum gray values after shrinkage.

Finally, perform Poisson fusion on the local remapped and originally merged connected regions. As a result, it increases the contrast and spatial details without changing the spectral information.

## 3. Results and Discussion

In the third chapter, the experiment is performed on the proposed algorithm and the experimental results are discussed. The environment and settings for the experiment are described in Section 3.1. The experimental datasets are then described in Section 3.2. The experimental results using the proposed algorithm are shown in Section 3.3. In Section 3.4, we perform subjective visual comparison analysis with traditional quality improvement algorithms and use five evaluation indicators for objective comparison. In Section 3.5, we classify quality improvement HSI and degraded HSI and further discuss the classification results.

### 3.1. Experimental Environment and Setting

A computer with 16 GB RAM and a 12th Gen Intel(R) Core (TM) i9-12900H 2.50 GHz processor running Windows 11 was used. The studies were performed in MATLAB R2022b. The Indian_Pines, PaviaU, and Salinas datasets were utilized for validation in this paper to estimate the effectiveness of the proposed algorithm.

### 3.2. Dataset Description

In order to prove the denoising and contrast enhancement performance of the proposed algorithm under degraded HIS, we add different Gaussian noises into three public hyperspectral remote sensing datasets (Indian_Pines, Pavia_University, and Salinas datasets) as experimental data. Meanwhile, to further prove the reliability of quality enhancement, we classify preprocessed HSI and unprocessed HSI.

The Indian Pines dataset [26] is a hyperspectral image of Indian pine trees in Indiana, USA, imaged by the airborne visible, infrared imaging spectrometer (AVIRIS) in 1992. Its size is 145 × 145. Wavelength range: 0.4–2.5 μm. Two hundred bands are the object of study [27], and the spatial resolution is about 20 m. During imaging, it is easy to be affected by atmospheric and other factors, resulting in noise interference and low dynamic range, which makes subsequent classification difficult. There are 16 types of ground objects, and the detailed category information is shown in Table 1.

The PaviaU dataset [28] is the hyperspectral data obtained by the German Reflective Optics Spectrographic Imaging System (ROSIS-03) in Pavia, Italy, in 2003. The spectral imager is sensitive to 0.43–0.86 μm, images 115 bands continuously, and the spatial resolution is 1.3 m. Among them, we eliminate 12 bands due to the influence of noise, remaining the 103 spectral bands. This data is 610 × 340, containing 42,776 object pixels and 164,624 background pixels. These object pixels contain nine ground objects: trees, asphalt roads, bricks, and meadows. The detailed category information is shown in Table 1.

The Salinas dataset [28] was also taken with an AVIRIS imaging spectrometer. It is an image of Salinas Valley in California, USA. Its spatial resolution reaches 3.7 m. After removing the 108–112, 154–167, and 224 wavebands affected by atmospheric water vapor [27], 204 waveband images remain. The image size is 512 × 217, so 111,104 pixels are included. Among them, 56,975 are background pixels, and 54,129 pixels can be applied to classification. These pixels are divided into 16 categories, such as fallow and celery. The detailed category information is shown in Table 1.

The three public datasets used for experiments are available at the website https://www.ehu.eus/ccwintco/index.php/Hyperspectral_Remote_Sensing_Scenes. (accessed on 10 August 2021).

### 3.3. Experimental Result of Quality Improvement on Three Public Datasets

First, we add Gaussian noise (σ = 5) to the Indian Pines dataset to generate degraded data. Then, we use the proposed algorithm to conduct quality improvement experiments. Figure 2 shows the spatial visualization results of seven bands selected from the Indian Pines dataset of 200 bands. Figure 3 shows the spectral curve.

The first row in Figure 2 is the original data of the Indian Pines dataset. The second row is the degraded HSI with Gaussian noise (σ = 5). The last row is the quality improvement result using the proposed algorithm.

The clean HSI can be recovered from degraded HSI in the third row using the proposed method. The second row shows that the degraded HSI is still in a low dynamic range. Then, by using the proposed algorithm, the contrast, spatial information, and noise interference have a noticeable improvement in visual effect. Interestingly, the original data contains noise for band one in the first row of Figure 2a. After adding Gaussian noise of five, the added and unknown noise in its data is removed, demonstrating that the proposed algorithm has good denoising performance.

Next, we add the same noise to each band. Because the brightness value of each band is different, there are varying degrees of noise interference in different bands. As shown in the second row of Figure 2 and Figure 3, due to the high average brightness of band ten, the added Gaussian noise of five belongs to the low noise. However, for band 170, due to its low average brightness, the added Gaussian noise of five belongs to relatively high noise, equivalent to introducing different levels of noise into the dataset. In this way no matter what, at a low noise level (band 10) or high noise level (band 170), data can be recovered well after quality improvement. Moreover, Figure 3 proves that the spectral curve of improved data is close to its true spectrum. It is confirmed that the proposed algorithm can not only complete the denoising and contrast enhancement, but also keep the spectral structure information well.

To perform quality improvement on the degraded PaviaU data, we add noise (σ = 5). As shown in Figure 4, we uniformly selected the original data, degraded images, and quality improvement images of bands 1, 10, 20, 30, 50, 80, and 100 for visual display. From the second row of Figure 4, the degraded images are visually disturbed by different degrees of noise. At the same time, both the original data, the first row of Figure 4, and the degraded data have low overall contrast. As a result, it is difficult to interpret the various targets in the data, which seriously affects the user’s ability to interpret the data. After conducting quality enhancement using the proposed algorithm, the visual effect, space contrast, and brightness are effectively improved in Band 1, Band 10, and Band 20. In these bands, compared with the degraded data, various targets can be seen, which is conducive to the interpretation of users. In band 80 and band 100, the overall brightness was improved compared with the front bands, but the noise interference was more serious. The proposed algorithm can effectively denoise and enhance the image’s contrast while preserving the image’s details and recover the object information. Figure 4 shows that the propose algorithm effectively improves the overall hyperspectral spatial quality.

Next, we further prove that the proposed algorithm does not affect the spectral information of ground objects while improving the spatial quality. We select the pixel (311,311) to extract the spectral information and display it in Figure 5. The blue curve is the spectral curve of the original data. The red curve is the degraded spectral curve. The orange curve is the quality-improved spectrum. The spectral profile after quality enhancement can effectively restore the spectral features of the ground target in each band. Zoomed in between band 70 and band 83, the improved spectral curve is very close to the real spectrum, consistent with the expected quality improvement.

We next perform quality enhancement on the degraded Salina dataset. As shown in Figure 6, we selected the experimental results of Salina data bands 1, 10, 40, 80, 120, 160, and 210 for visualization. The experimental results using the proposed algorithm obtain visual effects well. In band 1 (the first row of Figure 7a), the original data has striping noise in the transverse direction. The second row of Figure 7a adds Gaussian noise, and the effective information of this band is submerged in the mixed noise. Using the proposed algorithm, we remove the Gaussian and strip noise well, and recover the valid information. Band 40 data also typically shows a quality improvement. Not only is the noise of the data removed, but also the contrast and brightness of the spatial information are improved. The spectral information at position (151,151) is shown in Figure 7, and the spectral profile, after the quality enhancement, still maintains the inherent spectral properties of the target.

This section visualizes the degraded and quality-enhanced HSI results in terms of space and spectrum on three datasets, explaining that the proposed algorithm could improve HSI quality. Section 3.4 compares the proposed algorithm and related quality improvement algorithms, and analyzes them from subjective and objective aspects to demonstrate that the proposed algorithm has improved the overall results. Moreover, in Section 3.5, the quality improvement of HSI is applied to the classification task, further demonstrating the effectiveness of quality improvement for subsequent applications.

### 3.4. Evaluation of Quality Improvement Compared with Other Methods

Quality improvement aims at HSI with low contrast and noise interference. The classical HSI denoising algorithm is the BM4D algorithm, and the contrast enhancement includes linear stretching (LS) and histogram equalization (HE). Therefore, in order to further evaluate the proposed algorithm, the quality improvement algorithm is further discussed with BM4D [20], BM4D combined with linear stretching (BM4D + LS) [20,29], and BM4D combined with histogram equalization (BM4D + HE) [20,29].

We perform the relevant comparative experiments on three datasets and the visualization results are presented in Figure 8. Figure 8a shows that the degraded data of Indian Pines in band 50 has ambiguous targets and moderate noise interference. Although it removes the noise effectively, there is low contrast in it using the BM4D denoising algorithm. After utilizing the linear stretching on BM4D, the overall contrast is still low. There is a status quo with fog, as shown in the fourth column of Figure 8a, since the denoised pixels have pixels that are too bright or too dark, while other pixels are limited to a narrow histogram range. Moreover, it leads to low-quality visualizations. The comparison algorithms on the PaviaU and Salinas datasets also show similar performance. The overall brightness is low using the BM4D algorithm on these two datasets. After linear stretching, the data contrast is not improved because the maximum and minimum values of the denoised data limit the histogram distribution. Histogram equalization based on the BM4D algorithm improves the contrast, but the over-exposure phenomenon exists in local areas, leading to the loss of local information. Moreover, the use of linear stretching and histogram equalization for each band results in breaking the spectral characteristics of the ground truth, as shown in Figure 9. For BM4D + LS and BM4D + HE, although they improve the spatial contrast in some bands, the target spectral information is affected. In this way, it affects the subsequent spectral tasks, such as classification and recognition. The proposed quality improvement method not only improves the spatial visual effect, but also maintains the spectral characteristics target. Therefore, it is beneficial for subsequent HSI processing and other related tasks.

To compare these results, we use five indicators: Spectral Angle (SA) [10], Peak Signal to Noise Ratio (PSNR) [30], Structural Similarity (SSIM) [31], Brightness [30], and Contrast [30], to evaluate the experiment results.
(2)$$SA\left(x\right)=cos^{-1}\frac{d^{T}x}{{\left(d^{T}d\right)}^{1/2}{\left(x^{T}x\right)}^{1/2}},$$
where, *d* represents spectral vector of ground-truth, *x* represents spectral vector after quality improvement.
(3)$$SSIM\left(x,y\right)=\frac{\left(2\mu_{x}\mu_{y}+C_{1}\right)\left(2\sigma_{xy}+C_{1}\right)}{\left(\mu_{x}^{2}+\mu_{y}^{2}+C_{1}\right)\left(\sigma_{x}^{2}+\sigma_{y}^{2}+C_{2}\right)},$$
where, $\mu_{x}$ and $\mu_{y}$ represent mean value of the ground truth and the quality improvement HSI, $\sigma_{x}$ and $\sigma_{y}$ represent standard deviation of the ground truth and the quality improvement HSI. $\sigma_{xy}$ denotes covariance, and $C_{1}$ and $C_{1}$ are set as 0.0001 and 0.0009, respectively.
(4)$$PSNR=10\times \frac{log_{10}{\left(2^{n}-1\right)}^{2}}{MSE},$$
where, MSE represent mean square error. *n* represents digitalizing bit of HSI.
(5)$$Brightness=\sum_{l=0}^{2^{n}-1}l\times p\left(l\right),$$
(6)$$\;Contrast=\sqrt{\sum_{l=0}^{2^{n}-1}(l-Brightness\times p\left(l\right)},$$
where, l represents the gray value of HSI, p(l) represents the probability of gray level l.

Table 2 shows the results of five indicators on three public datasets, and Figure 10 shows the index variation diagram of each band in the Indian Pines dataset. The SA indicator on the Indian Pines dataset dropped from 9.740 to 2.891, confirming the stability of spectral information during quality improvement. Meanwhile, its spectral information is closer to the actual spectrum, proving that the quality improvement could provide a guarantee for subsequent classification tasks.

Similarly, the contrast index increased from 0.0345 to 0.0918, and the brightness index increased from 0.1963 to 0.2670. It can be seen from Figure 10d that the index of each band after enhancement is also higher than the ground truth. We prove the effectiveness of the histogram redistribution algorithm in Section 2.2. SSIM was improved from 0.4725 to 0.9408, proving that the proposed method recovered spatial structure information.

### 3.5. Applied to Classification Tasks

In order to show the influence of hyperspectral quality improvement on classification, we send quality improvement HSI and degraded HSI from three public datasets into the 3dCNN classification model [32].

The network model comprises four 3D convolutional layers with kernel sizes of 3 × 3 × 3, 1 × 1 × 3, and 1 × 1 × 2. ReLU is used as the activation function. The final output is through the full connection layer. The setting patch size is 7, and epoch is 150. The initial learning rate is 0.001. In order to further verify that quality enhancement can improve the classification accuracy under different conditions, we use multiple data partition ratios for multiple verifications. The dividing ratio of training and test data is 50%, 25%, and 15%.

The average Kappa and classification accuracy of the ten experiments on three datasets are shown in Table 3. After using the proposed algorithm on three datasets, its classification accuracy is higher than degraded data. Specifically, the average classification accuracy of the quality enhanced by the proposed algorithm is 96.318%, 93.886%, and 89.556% on the three types of dividing ratios, respectively. Compared with the dataset with degraded HSI, the average classification accuracy is improved by 4.12%, 6.81%, and 13.93%, respectively. The Kappa coefficient improved under different partition ratios.

Table 4, Table 5 and Table 6 show the classification accuracy of 16 categories in the Indian Pines dataset on a 50% dividing ratio, eight categories in the PaviaU dataset, and 16 categories in the Salina dataset, respectively. In each category, the data after quality improvement is improved compared with the degraded data.

Figure 11, Figure 12 and Figure 13 show the final classification results. The noise points of the image with improved quality in each category are significantly less than the classification results of degraded data. It also proves the effectiveness of data quality improvement for subsequent applications.

## 4. Conclusions

In this paper, a hyperspectral image quality improvement algorithm based on texture search and histogram redistribution is proposed. Firstly, a new clustering strategy for searching along edge texture is proposed in 4D block clustering for denoising. Then, after the secondary polymerization, contrast enhancement was performed by histogram redistribution and Poisson fusion. There are two main advantages. On the one hand, we use the texture search strategy to make 3D block aggregation sparser to improve the accuracy of the four-dimensional transformation. On the other hand, we exploit the histogram redistribution method to stabilize spectral information and enhance spatial contrast. Experimental results show that the final quality enhancement is significant, and the spectral information is stably retained. More importantly, it effectively improves the accuracy of subsequent classification tasks.

It is a preliminary attempt to improve HSI quality by combining denoising and spatial contrast enhancement. However, the spatial spectrum has not been improved in some bands. In the future, more robust quality improvement algorithms to further improve HSI quality should be considered. For example, to restore more information about the local spatial contrast, and to improve spectral accuracy by learning spectral reflectance properties.

## Figures and Tables

**Figure 1 sensors-23-02731-f001:**
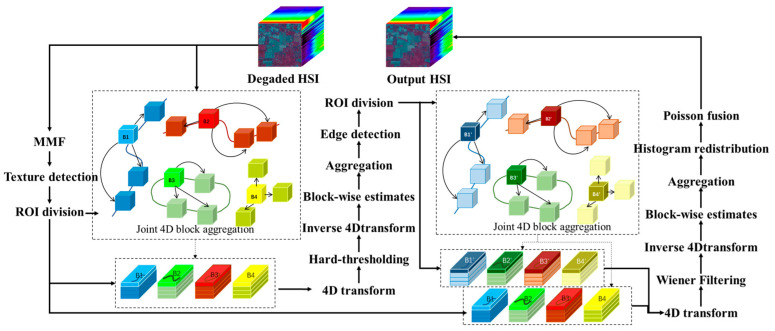
Process flow of the proposed algorithm. The function and process of each block are detailed in Section 2.

**Figure 2 sensors-23-02731-f002:**
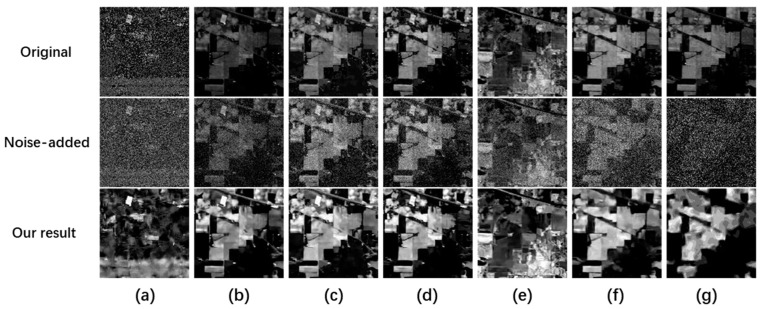
Visualization results of the Indian Pines dataset at different bands. The first row represents the raw grayscale. The second row represents the noise map with added noise. The third row represents the quality improvement result. Each column from left to right represents (**a**) band 1, (**b**) band 10, (**c**) band 20, (**d**) band 30, (**e**) band 50, (**f**) band 120, and (**g**) band 170.

**Figure 3 sensors-23-02731-f003:**
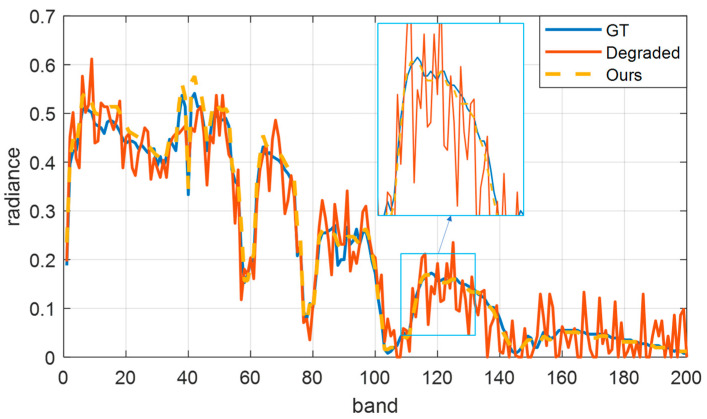
Spectral curve at (40,40) in the Indian Pines dataset. Blue represents the original spectral curve, red represents the spectral curve of degraded data, and yellow represents the spectral curve after quality improvement.

**Figure 4 sensors-23-02731-f004:**
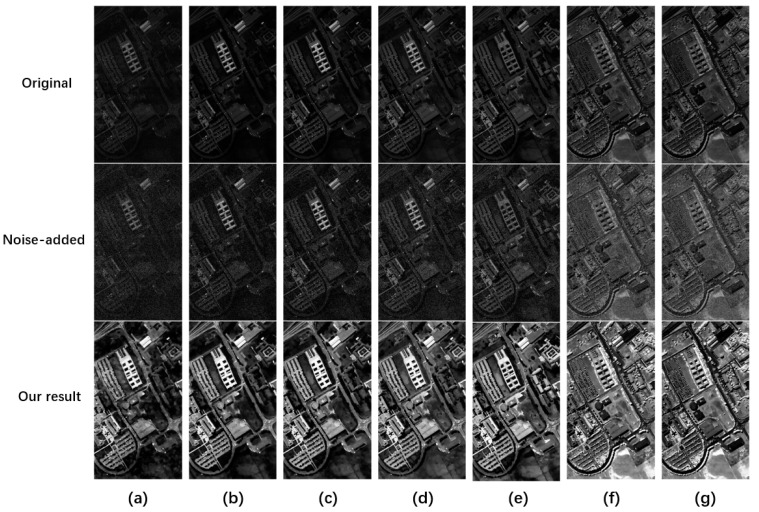
Visualization results of the Pavia University dataset at different bands. The first row represents the original data. The second row represents the degraded map with added noise (σ = 5). The third row represents the quality improvement result. Each column from left to right represents (**a**) band 1, (**b**) band 10, (**c**) band 20, (**d**) band 30, (**e**) band 50, (**f**) band 80, and (**g**) band 100.

**Figure 5 sensors-23-02731-f005:**
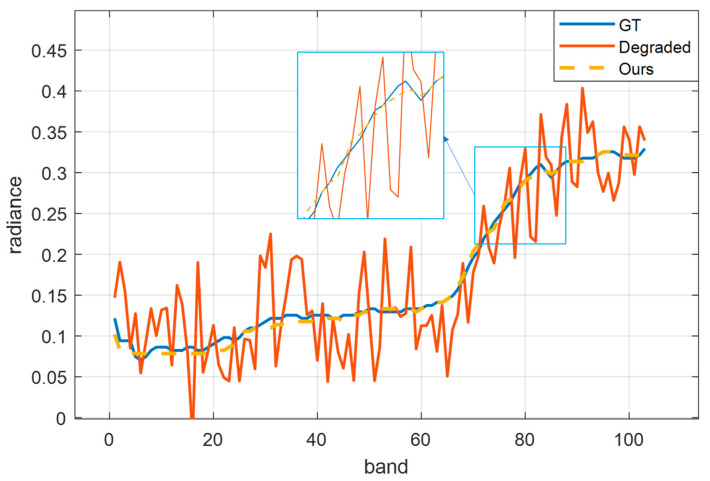
The spectral curve at (311,311) in the PaviaU dataset. Blue represents the original spectral curve, red represents the spectral curve of degraded data, and yellow represents the spectral curve after quality improvement.

**Figure 6 sensors-23-02731-f006:**
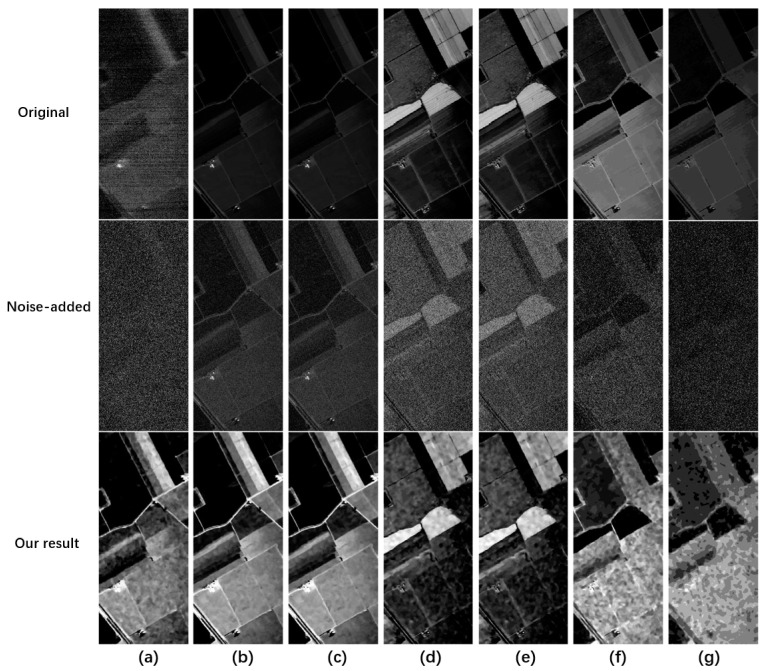
Visualization results of the Salinas dataset at different bands. The first row represents the original data. The second row represents the noise map with added noise (σ = 10). The third row represents the quality improvement result. Each column from left to right represents (**a**) band 1, (**b**) band 10, (**c**) band 40, (**d**) band 80, (**e**) band 120, (**f**) band 160, and (**g**) band 210.

**Figure 7 sensors-23-02731-f007:**
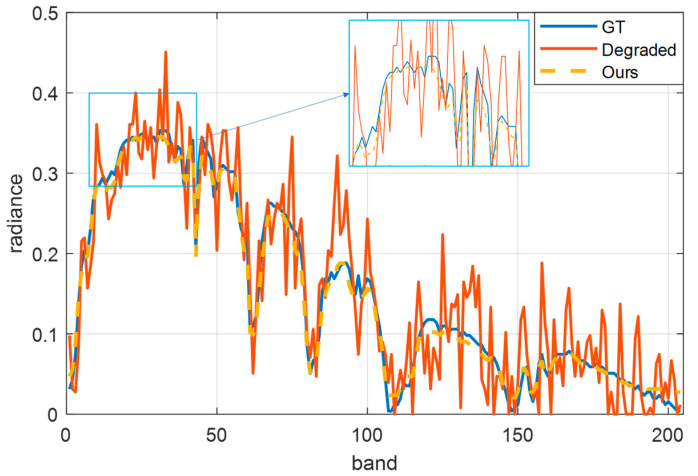
Spectral curve at (151,151) in the Salinas dataset. Blue represents the original spectral curve, red represents the degraded spectral curve, and yellow represents the spectral curve after quality improvement.

**Figure 8 sensors-23-02731-f008:**
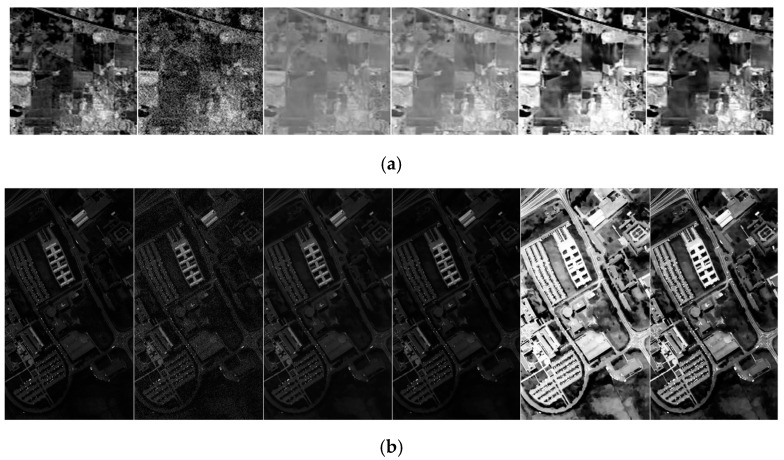
Visualization comparison results of three public datasets using different methods. (**a**) Experimental results of band 50 in the Indian Pines dataset. (**b**) Experimental results of band 1 in the PaviaU dataset. (**c**) Experimental results of Band 11 in the Salinas dataset. Each line from left to right represents ground truth, degraded data, BM4D, BM4D + LS, BM4D + HE, and the proposed method.

**Figure 9 sensors-23-02731-f009:**
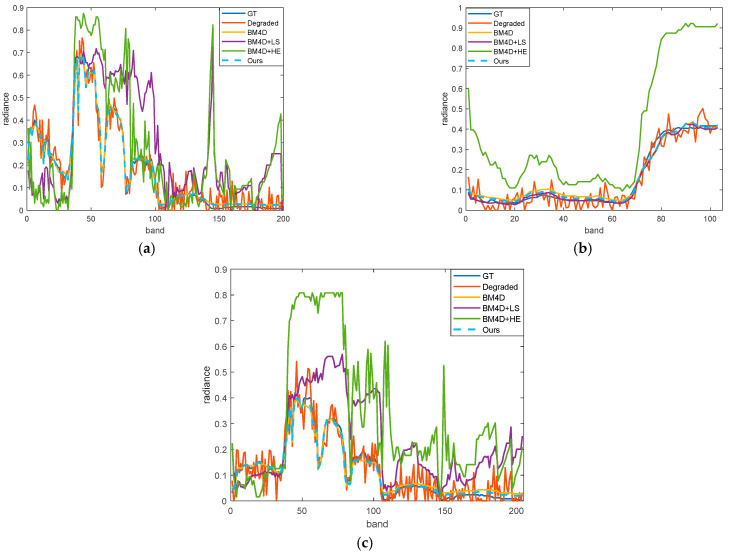
The spectral curve results of different comparison algorithms. Blue represents the original spectral curve, red represents the spectral curve of degraded data, yellow represents the curve results using the BM4D algorithm, purple represents the curve results using the BM4D + LS algorithm, green represents the curve results using the BM4D + HE algorithm, and blue represents the spectral curve after quality improvement. (**a**) The spectral curve at (111,111) in the Indian Pines dataset. (**b**) The spectral curve at (171,171) in the PaviaU dataset. (**c**) The spectral curve at (101,101) in the Salinas dataset.

**Figure 10 sensors-23-02731-f010:**
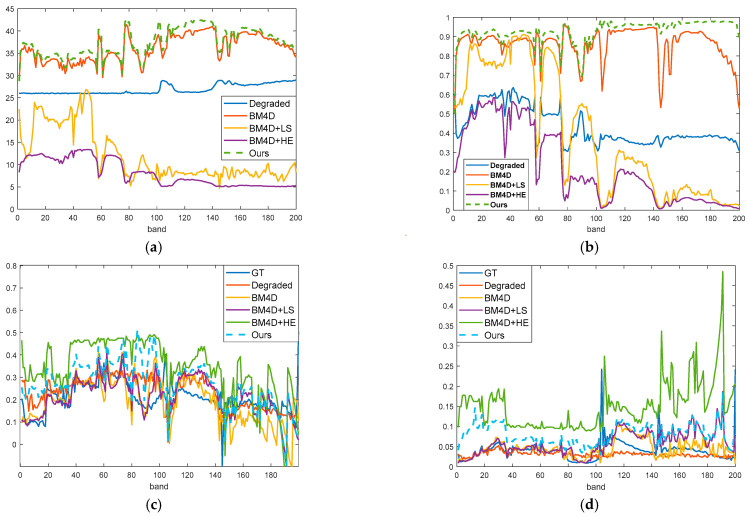
Index curves for each band of the Indian Pines dataset. The blue, red, yellow, purple, and dotted green represent each band’s index with degraded data, BM4D, BM4D + LS, BM4D + HE and proposed results, respectively. (**a**) PSNR evaluation. (**b**) SSIM evaluation. (**c**) Brightness evaluation. (**d**) Contrast evaluation.

**Figure 11 sensors-23-02731-f011:**
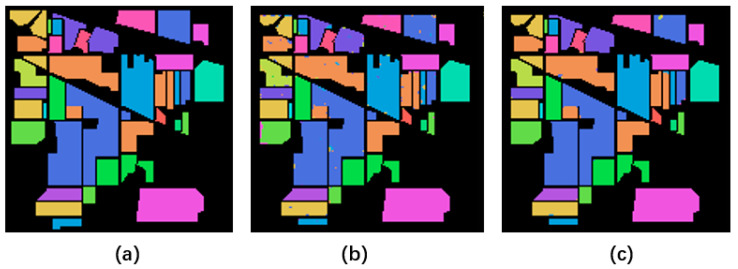
Classification results on the Indian Pines dataset. (**a**–**c**) represent category labels of ground truth, classification result on degraded data, and classification result after quality enhancement, respectively.

**Figure 12 sensors-23-02731-f012:**
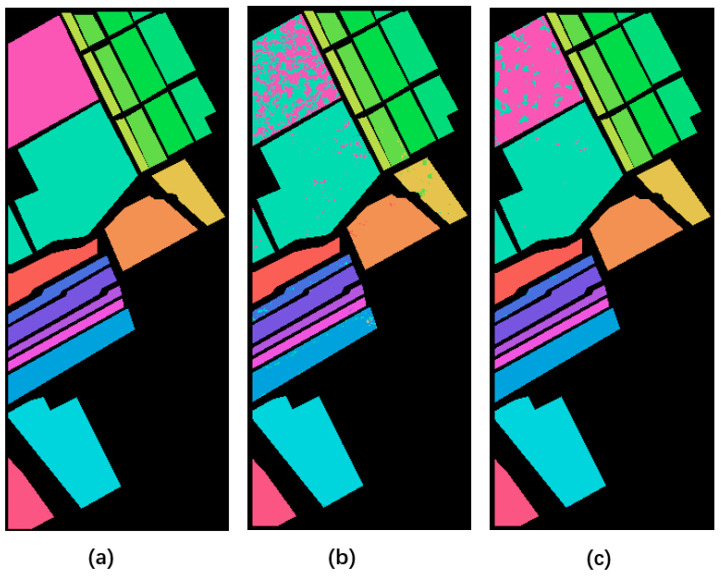
Classification results on the PaviaU dataset. (**a**–**c**) represent category labels of ground truth, classification result on degraded data, and classification result after quality enhancement, respectively.

**Figure 13 sensors-23-02731-f013:**
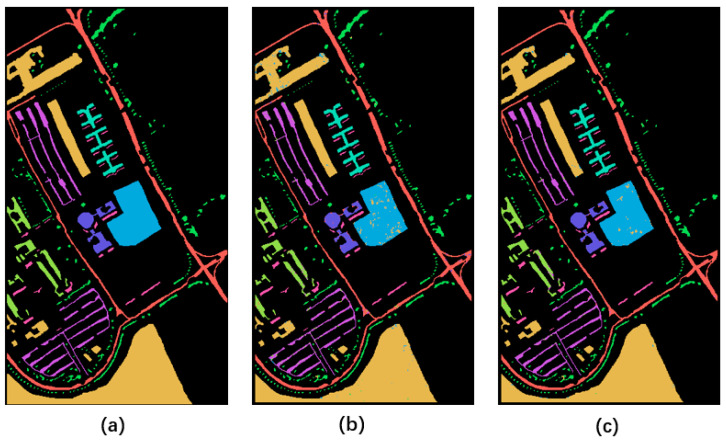
Classification results on the Salinas dataset. (**a**–**c**) represent category labels of ground truth, classification result on degraded data, and classification result after quality enhancement, respectively.

**Table 1 sensors-23-02731-t001:** The results of five indicators on three pubic datasets.

Class	Indian Pines	Salinas	PaviaU
C1	Alfalfa	Brocoli_green_weeds_1	Meadows
C2	Corn-notill	Brocoli_green_weeds_2	Gravel
C3	Corn-mintill	Fallow	Trees
C4	Corn	Fallow_rough_plow	Painted metal sheets
C5	Grass-pasture	Fallow_smooth	Bare Soil
C6	Grass-trees	Stubble	Bitumen
C7	Grass-pasture-mowed	Celery	Self-Blocking Bricks
C8	Hay-windrowed	Grapes_untrained	Shadows
C9	Oats	Soil_vinyard_develop	Meadows
C10	Soybean-notill	Corn_senesced_green_weeds	-
C11	Soybean-mintill	Lettuce_romaine_4wk	-
C12	Soybean-clean	Lettuce_romaine_5wk	-
C13	Wheat	Lettuce_romaine_6wk	-
C14	Woods	Lettuce_romaine_7wk	-
C15	Buildings-Grass-Trees-Drives	Vinyard_untrained	-
C16	Stone-Steel-Towers	Vinyard_vertical_trellis	-

**Table 2 sensors-23-02731-t002:** The quantitative results of three pubic datasets. (Bold texts indicate the best performance).

		SA	PSNR	SSIM	Brightness	Contrast
Indian Pines	GT	-	-	-	0.1963	0.0339
	degenerate	9.7401	26.7679	0.4725	0.2004	0.0345
	BM4D	3.3897	35.2064	0.8775	0.2002	0.0321
	BM4D + LS	39.9986	9.6317	0.3205	0.4282	0.0451
	BM4D + HE	43.4669	7.2456	0.2092	**0.4998**	0.0834
	Proposed algorithm	**2.8914**	**38.3781**	**0.9408**	0.2670	**0.0918**
PaviaU	GT	-	-	-	0.1736	0.0125
	degenerate	15.561	26.2975	0.4868	0.1748	0.0146
	BM4D	4.0191	30.2303	0.9295	0.1746	0.0121
	BM4D + LS	4.4843	36.0436	0.9364	0.1651	0.0127
	BM4D + HE	34.4843	8.00190	0.3947	**0.4999**	**0.0860**
	Proposed algorithm	**3.6901**	**38.6992**	**0.9432**	0.3506	0.0603
Salinas	GT	-	-	-	0.1310	0.0142
	degenerate	15.4885	26.7626	0.3911	0.1350	0.0154
	BM4D	4.0345	33.4485	0.9214	0.1347	0.0134
	BM4D + LS	37.1998	12.7385	0.4668	0.3017	0.0288
	BM4D + HE	40.5397	6.73277	0.2363	**0.5000**	**0.0845**
	Proposed algorithm	**3.1011**	**40.0777**	**0.9391**	0.3151	0.0753

**Table 3 sensors-23-02731-t003:** The classification results on different dividing ratios of three public datasets.

Datasets		50% (%)		25% (%)		15% (%)	
		Accuracy	Kappa	Accuracy	Kappa	Accuracy	Kappa
Indian Pines	Degenerate	93.776	92.9	81.865	79.3	72.922	68.9
	Proposed algorithm	96.702	96.3	94.224	93.4	86.111	84.1
Salinas	Degenerate	90.120	89.0	89.342	88.5	76.720	74.5
	Proposed algorithm	96.586	96.2	92.815	91.9	89.600	88.4
PaviaU	Degenerate	93.651	91.7	92.470	90.8	86.176	81.5
	Proposed algorithm	95.666	94.3	94.620	93.0	92.954	91.7
Average	Degenerate	92.516	91.2	87.892	86.2	78.606	75.0
	Proposed algorithm	**96.318**	**95.6**	**93.886**	**92.7**	**89.556**	**88.1**

**Table 4 sensors-23-02731-t004:** The classification accuracy of the 16 categories on the Indian Pines dataset.

Accuracy (%)	C1	C2	C3	C4	C5	C6	C7	C8
Degenerate	97.8	95.6	89.3	91.8	93.2	99.2	100	99.8
Proposed algorithm	100	98.7	94.3	97.5	96.6	99.9	96.3	99.8
**Accuracy (%)**	**C9**	**C10**	**C11**	**C12**	**C13**	**C14**	**C15**	**C16**
Degenerate	94.7	92.8	94.9	93	99.5	98.5	88.3	98.9
Proposed algorithm	100	98.4	97.9	98.8	99.5	99.9	89.1	96.7

**Table 5 sensors-23-02731-t005:** The classification accuracy of the 9 categories on the PaviaU dataset.

Accuracy (%)	C1	C2	C3	C4	C5	C6	C7	C8	C9
Degenerate	96.3	95.2	88.9	98.1	100	92.9	92.3	93.5	100
Proposed algorithm	97.6	96.7	92.0	98.7	100	97.5	97.2	95.9	99.7

**Table 6 sensors-23-02731-t006:** The classification accuracy of the 16 categories on the Salinas dataset.

Accuracy (%)	C1	C2	C3	C4	C5	C6	C7	C8
Degenerate	97.7	99.5	95.7	99.8	97.3	99.5	99.6	83.6
Proposed algorithm	98.4	100	99.9	99.9	99.9	99.6	99.7	95.2
**Accuracy (%)**	**C9**	**C10**	**C11**	**C12**	**C13**	**C14**	**C15**	**C16**
Degenerate	98.7	95.9	92.3	98.6	98.0	98.3	60.6	94.9
Proposed algorithm	99.8	98.4	96.5	99.0	99.2	99.0	91.1	94.1

## Data Availability

The data presented in this study are available on request from the corresponding authors.

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
