# Peer review of "Joint Texture Search and Histogram Redistribution for Hyperspectral Image Quality Improvement"

_sensors, 2023, doi:10.3390/s23052731_

Round 1

Reviewer 1 Report

English should be improved to make a clear presentation of this paper.

Avoid using "etc","we" or "our" in this paper.

Ours can be replace with the proposed algorithm.

Line 33 - "In this way" shall be replace with "consequently"

Line 52 - 58 - rephrase the sentences. Suggest - The experiment results and discussion are presented and discussed in section 3 and 4.

Figure 1 is not look like a flow chart. It is like process flow of the proposed algorithm. Caption in Figure 1 should be rewords. Where is section II?

Line 122 - Further elaborate the purpose of "to do local mapping"

Line 124 - "In this way" shall be replaced with "As a result"

In section 3.3 - "Line" shall be replace with "row". Please use consistently term. Line and row are used inconsistently in this section

Line 222- proposed scheme or proposed algorithm?

Line 298 - 303 - Suggest to rewords with "The proposed quality improvement method not only improves the spatial visual effect, but also maintains the spectral characteristics target. Therefore, it is beneficial for subsequent  processing of hyperspectral data and other related tasks"

Line 331 - Justify why the distribution dataset is 50% is used.

Perhaps, in the results, please justify why the proposed algorithm has improved the overall results.

Author Response

Thank you for the reviewers’ comments concerning our manuscript entitled “Joint texture search and histogram redistribution for hyper-spectral image quality improvement” (ID: sensors- 2241960). Those comments are all valuable and very helpful for revising and improving our paper, as well as the important guiding significance to our research. We have studied comments carefully and have made corrections which we hope meet with approval. Revised portions are marked in red on the paper. The revised manuscript is in the attachment.

We appreciate for Editors/Reviewers’ warm work earnestly and hope that the correction will meet with approval. If you think the modification is not good enough, we will continue to modify it. Once again, thank you very much for your comments and suggestions.

Reviewer 2 Report

Review comments on “Joint texture search and histogram redistribution for hyperspectral image quality improvement” by Hu et al. submitted to Sensors.

This paper proposed a novel idea to improve the quality of hyper-spectral image quality based on texture search and histogram redistribution. It has significant contribution to the field of HSI quality improvement, which is interesting. The motivation is right for the current HSI quality improvement research, and the analysis is adequate. A minor revision is needed.

1. In the introduction, the innovation of the manuscript should be introduced in detail.

2. Please give more details about the evaluation criteria, best with formulation.

3. In the conclusion part, some potential future work should be discussed.

Author Response

Thank you for the reviewers’ comments concerning our manuscript entitled “Joint texture search and histogram redistribution for hyper-spectral image quality improvement” (ID: sensors-2241960). Those comments are all valuable and very helpful for revising and improving our paper, as well as the important guiding significance to our research. We have studied comments carefully and have made corrections which we hope meet with approval. Revised portions are marked in red on the paper. The revised manuscript is in the attachment.

We appreciate for Editors/Reviewers’ warm work earnestly and hope that the correction will meet with approval. If you think the modification is not good enough, we will continue to modify it. Once again, thank you very much for your comments and suggestions.
